# Impact of the COVID-19 Pandemic on Neonatal Nursing Practicum and Extended Reality Simulation Training Needs: A Descriptive and Cross-Sectional Study

**DOI:** 10.3390/ijerph20010344

**Published:** 2022-12-26

**Authors:** Sun-Yi Yang

**Affiliations:** College of Nursing, Daejeon Medical Campus, Konyang University, 158, Gwanjeodong-ro, Seo-gu, Daejeon 35365, Republic of Korea; lavender799@gmail.com; Tel.: +82-42-600-8560

**Keywords:** nursing students, nursing, intensive care, neonatal, clinical practice, patient simulation

## Abstract

This study investigated the neonatal intensive care unit (NICU) clinical practicum status during the COVID-19 pandemic and the need for extended reality (XR)-based training for neonatal care. A structured questionnaire was distributed to 132 prelicensing nursing students. Data were analyzed using importance-performance analysis and Borich needs analysis. Students wanted to use XR to learn about treating high-risk preterm infants. COVID-19 limited clinical training in NICUs, and most students preferred training in XR programs to improve their nursing competency for neonates. There is a large demand for nursing skills concerning high-risk newborns and hands-off training.

## 1. Introduction

Clinical practicum in the Bachelor of Science in Nursing curricula aims to enhance nursing competency in undergraduates through direct and indirect incorporation of their nursing knowledge and theories in clinical practice [1]. However, students’ engagement in direct patient care is generally prohibited in clinical settings for reasons such as patient safety and prevention of medical malpractice; thus, clinical practicums predominantly comprise observation and explanations [2]. Even this limited clinical exposure, particularly in neonatal intensive care units (NICUs), has been restricted or completely suspended during the unprecedented coronavirus disease 2019 (COVID-19) pandemic to prevent the spread of infection to neonates [3,4,5]. Thus, nursing schools had to quickly transition from clinical practicum to remote and online methods [6,7]. Owing to the continued emergence of COVID-19 variants, the return to normal clinical practicum remains a challenge [8].

The provision of neonatal care to high-risk neonates in the NICU through inadequately trained nurses can threaten neonatal health and safety and ultimately increase neonatal mortality and morbidity rates [9,10]. Therefore, devising training measures to substitute for on-site NICU clinical practicum is crucial [11,12]. A rise in the use of simulation-based education (SBE) to address the drawbacks of clinical practice and cultivate clinical competency in students has been observed [13]. SBE facilitates the improvement of nursing students’ critical clinical performance as it permits the repeated practice of clinical skills, without threatening patient safety, in a setting resembling actual clinical environments [14]. Huang et al. [15] reported that high-fidelity SBE markedly improved high-risk neonatal nursing competency.

However, the high cost of the simulator and workforce is a major barrier to face-to-face SBE using a high-fidelity simulator, and adequate training is hindered by a shortage of simulation equipment and an inability to accommodate all students [16]. Extended reality (XR) simulation training—utilizing virtual, augmented, and mixed reality—has been developed to address these issues and is gaining traction [17,18,19].

Prior to developing an XR simulation training program, it is necessary to examine nursing competence and its associated factors among nursing students who experienced restrictions in clinical practicum owing to COVID-19. The competence of prospective nursing school graduates influences their clinical performance as nurses, and the clinical nursing competency of final-year nursing students is directly linked to the quality of care provided for patients and patient safety [20]. A mixed-methods study that identified the predictors of learning flow in nursing students during the COVID-19 pandemic [21] reported that nursing students were concerned that they might not have been adequately equipped owing to clinical practicum restrictions. Hence, it is essential to examine nursing students’ clinical competency and identify associated factors to establish systematic practicum systems and strategies [22,23]. Park and Ji [24] emphasized surveying nursing students’ needs for neonatal nursing education to boost the effectiveness of neonatal nursing education. Students’ level of interest in the topic of learning in an e-learning environment has a considerable impact on their learning outcomes [25,26].

Although handoff communication competency is an essential element in ensuring continued and accurate nursing and patient safety, it is often excluded from the essential nursing curricula [27]. Therefore, new nurses continue to experience handoff errors. Unorganized nurse handoffs without an established system cause confusion and can harm patients’ health [28]. In a qualitative needs assessment of NICU training, Koo and Lee [4] reported that students needed substantial handoff observation and practice. Therefore, providing systematic handoff training through undergraduate nursing education to enhance the ability of nursing students to accurately communicate among medical staff is necessary [29].

This study examined the effects of the COVID-19 pandemic on neonatal nursing clinical practicum and undergraduate nursing students’ training needs. The results will serve as foundational data for the development of an XR-based NICU nursing simulation training program.

## 2. Materials and Methods

### 2.1. Theoretical Framework

The need for educational and training programs that use digital technology for continuous medical personnel training in the field of pediatrics has been emphasized. Further, a systematic review and meta-analysis confirmed that digital education was effective enough to replace traditional medical personnel training [17]. The development of augmented reality technology—including virtual and augmented reality—which enables human movement, interaction, and experience, has been accelerated, and the need for it to be introduced into clinical training has been raised [30]. Moreover, using XR in clinical training is advantageous because it is not only as effective as traditional training methods in terms of education, but also because the enjoyment of learning and motivation have been highlighted. Further, as one of the next-generation medical education methods, the use of XR in clinical training is receiving a lot of attention [31]. Accordingly, as clinical training using XR technology is likely to become an essential component of clinical education, it is necessary to identify the needs of learners with regard to the kind of programs that they want to be developed using XR technology in the specialized field.

### 2.2. Study Design and Purpose

This study employed a descriptive, cross-sectional design.

First, I examined newborn and high-risk neonatal nursing competency, NICU training needs and experience, and the training needs for nursing care handoffs among undergraduate nursing students. Second, I analyzed the differences in newborn and high-risk neonatal nursing competency, NICU training needs and experience, and training needs for nursing care handoffs to families per the general characteristics of prelicensure nursing students. Third, I determined the variances in NICU training needs and experiences of prelicensure nursing students using importance-performance analysis (IPA). Fourth, I identified differences in educational needs for neonatal nursing and care transitions to family per prior handoff training among prelicensure nursing students.

### 2.3. Participants

The participants were prelicensure nursing students enrolled in one of the 116 Bachelor of Nursing programs across South Korea. The inclusion criteria were fourth-year nursing students who had completed a pediatric nursing practicum. The exclusion criteria were clinical work experience prior to nursing program admission and prior participation in an extracurricular simulation training program, as these experiences were considered likely to influence the results.

The sample size was calculated using G*power 3.1.9 software [32]. For a paired t-test to analyze differences in NICU training needs and experiences with an effect size (d) of 0.25, significance level (α) of 0.05, and power (1 − β) of 0.80, the minimum sample size was 128. Referring to previous studies [33,34] that reported a withdrawal rate of 1.7–2.1%, I anticipated a 3% dropout rate, and thus, sampled 132 prelicensure nursing students through convenience sampling. Since all 132 prelicensure nursing students answered all the questions, they were all included in the final analysis.

### 2.4. Instruments

#### 2.4.1. Neonatal Nursing Performance

Neonatal nursing performance, developed and validated by Park [35], was used. This tool comprises 25 items of physical assessment performed upon NICU admission, rated on a four-point Likert scale ranging from 1 (“strongly disagree”) to 4 (“strongly agree”). The total score ranges from 25 to 100, with higher scores indicating higher neonatal nursing performance. Subscale scores were calculated as the average item score and presented as a value ranging from 1–4. Cronbach’s α was 0.92 at the time of development and 0.95 in this study.

#### 2.4.2. High-Risk Neonatal Nursing Performance

The High-risk Neonatal Nursing Performance Scale, validated and converted to a self-report scale by Park and Ji [24] based on the High-risk Neonatal Severity Scale developed and validated by Kim, Moon, Kim, and Sim [36], was used. Comprising 39 items rated on a four-point Likert scale ranging from 1 (“strongly disagree”) to 4 (“strongly agree”), the total score ranges from 39 to 156, with higher scores indicating higher high-risk neonatal nursing performance. Subscale scores were calculated as the average item score and presented as a value ranging from 1–4. Cronbach’s α was 0.90 at the time of development and 0.97 in this study. 

#### 2.4.3. NICU Training Needs

The Neonatal Nursing Clinical Practicum Experience Scale developed and validated by Park and Ji [24] was used after changing the answer choices for a self-report assessment of NICU training needs. Each item is rated on a four-point Likert scale ranging from 1 (“not needed at all”) to 4 (“needed very much”). The total score ranged from 21 to 84, with a higher score indicating greater simulation training needs for the topic. Subscale scores were calculated as the average item score and presented as a value ranging from 1–4. Cronbach’s α was 0.95 at the time of development and 0.96 in this study.

#### 2.4.4. NICU Training Experience

The Neonatal Nursing Clinical Practicum Experience Scale developed and validated by Park and Ji [24] was used. Comprising 21 items rated on a four-point Likert scale ranging from 1 (“no experience at all”) to 4 (“adequate experience”), the total score ranges from 21 to 84, with higher scores indicating students perceived to have adequate NICU clinical training. Domain and subscale scores were calculated as the average item score and presented as a value ranging from 1–4. Cronbach’s α was 0.95 at the time of development and 0.96 in this study.

#### 2.4.5. Neonatal and Family Handoff Training Needs

The Neonatal and Family Handoff Training Needs Scale, developed based on the Pediatric Nursing Handoff Needs Scale developed and validated by Park et al. [34] and Nursing Diagnoses 2021–2023 [37], was used after modifying and validating the tool for nursing students. This 46-item tool comprises 10 domains, with each item rated on a four-point Likert scale ranging from 1 (“very irrelevant”) to 4 (“very relevant”). A higher score indicates a greater perceived need for education and educational needs. Domain scores were calculated as the average item score and presented as a value ranging from 1–4. Cronbach’s α was 0.96 previously [34] and 0.97 in this study.

#### 2.4.6. General Characteristics

To understand the general characteristics of the participants, details regarding their sex, satisfaction with nursing major, satisfaction with clinical practice, newborn, and high-risk neonatal care training sites, secure a private university hospital, simulation training experience pertaining to neonatal care, handover training experience, demand for XR-based training for neonatal care, neonatal nursing performance, high-risk neonatal nursing performance, need for newborn intensive care training, newborn intensive care training experience, and the necessity for handover training-related contents were collected.

### 2.5. Data Collection

The purpose, method, participant protection, and questionnaire of the study were reviewed and approved by the Institutional Review Board of Konyang University (no. KYU-2022-04-012-001). Data were collected through a recruitment announcement linked to the study questionnaire, and posted on a popular website for nursing students and the social media of nursing school student councils in South Korea. The announcement specified that privacy and confidentiality were guaranteed, that participation was voluntary, and that participants could withdraw from the study at any time without any disadvantages. The participants were also informed that the study data would be used solely for research purposes and would remain anonymous and confidential. The online questionnaire was distributed as a link and took approximately 30 min to complete.

### 2.6. Data Analysis

Data were analyzed using PASW 27.0, for Windows (SPSS Inc., Chicago, IL, USA). Normality was tested using the Shapiro-Wilk test, and homoscedasticity was confirmed using the Levene test; hence, parametric statistical analyses were performed. The general characteristics of participants were analyzed using frequency, percentage, mean, and standard deviation. Differences in high-risk neonatal nursing performance, NICU training needs and experience, and neonatal and family handoff training needs per general characteristics were analyzed using mean and standard deviation, chi-square test, Fisher’s exact test, independent t-test, or analysis of variance. The correlations between NICU training needs and experiences were analyzed with mean and standard deviation, paired t-test, IPA [38], and Borich Needs Analysis [39]. The differences in neonatal and family handoff training needs per the presence or absence of handoff training experience were analyzed using mean and standard deviation and independent *t*-tests.

## 3. Results

### 3.1. Participants’ General Characteristics

The study sample consisted of a higher percentage of female students, most of whom were satisfied with their current major. More students had their neonatal nursing clinical practicum off-site as compared to on-site. Most did not attend a school with an affiliated hospital. Nearly two-thirds had prior neonatal nursing simulation training, but most had not received handoff training. Nearly all the students stated that XR-based (e.g., virtual, augmented, and mixed reality) neonatal nursing training programs were needed (Table 1).

### 3.2. Differences in Neonatal and High-Risk Neonatal Nursing Performance, NICU Training Needs and Experience, and Neonatal and Family Handoff Training Needs per General Characteristics

Table 2 presents differences in neonatal and high-risk neonatal nursing performance, NICU training needs and experience, and neonatal and family handoff training needs per participants’ general characteristics.

Neonatal and family handoff training needs were significantly higher among students who were highly satisfied with their major than moderately satisfied or dissatisfied with their major (F = 3.61, *p* = 0.030). High-risk neonatal nursing performance was significantly higher among students who were moderately satisfied with their clinical practicum than dissatisfied (F = 3.01, *p* = 0.043). Neonatal nursing performance (t = 9.39, *p <* 0.001) and NICU training experience (t = 9.03, *p <* 0.001) were significantly higher among students who had an on-site clinical practicum than an off-site clinical practicum. NICU training experience (t = 2.53, *p* = 0.012) was significantly higher among students in schools with an affiliated hospital. Neonatal nursing performance (t = 4.17, *p* = 0.006) was significantly lower among students who perceived the need for an XR-based neonatal nursing training program than among those who did not. Neonatal and family handoff training needs (t = 2.07, *p* = 0.041) were significantly higher among students who had prior handoff training.

### 3.3. Differences between NICU Training Needs and Experience

Table 3 shows the differences in NICU training needs and experiences. The mean NICU training needs score was 3.67 ± 0.50 out of 4. By domain, the score was highest for high-risk newborn care, followed by newborn care and newborn assessment. The top five topics reported for training needs were phototherapy, body temperature maintenance, infection prevention, oxygen therapy, and infusion therapy.

The mean NICU training needs score was 2.04 ± 0.79 out of 4. By domain, the score was highest for newborn assessment, followed by newborn care and high-risk newborn care. The five least previously trained topics were body temperature maintenance, respiration maintenance, neonatal behavioral status assessment, Apgar scoring, and nutrition supply.

I analyzed the differences between NICU training needs and experiences and found a gap between needs and experience in all 21 categories. The top five categories with the greatest gap between need and experience were gavage tube feeding, oxygen therapy, fluid therapy, attachment promotion, and incubator management.

In the Borich analysis, the training needs were highest for gavage tube feeding, followed by the oxygen therapy, fluid therapy, infection prevention, attachment promotion, incubator management, phototherapy, and maintenance of body temperature. 

In the IPA, the categories rated as “concentrate here” were oxygen therapy, maintenance of body temperature, infection prevention, gavage tube feeding, incubator management, phototherapy, fluid therapy, and attachment promotion in the high-risk newborn care domain (Figure 1).

### 3.4. Neonatal and Family Handoff Training Needs per Handoff Training

Table 4 shows the differences in neonatal and family handoff training needs per the handoff training. Students with prior handoff training showed higher training needs for constipation (t = 2.29, *p* = 0.023), deficient knowledge (t = 2.56, *p* = 0.011), hopelessness (t = 2.93, *p* = 0.004), anxiety (t = 2.70, *p* = 0.008), death anxiety (t = 3.04, *p* = 0.003), fear (t = 2.68, *p* = 0.008), powerlessness (t = 3.28, *p* = 0.001), and impaired resilience (t = 3.20, *p* = 0.002) than students without prior handoff training.

Neonatal and family handoff training needs were analyzed based on 10 domains and 46 nursing diagnoses; students showed the highest handoff training needs for domain eight (safety/protection). Nursing diagnoses with the highest training needs reported in each domain were a risk for imbalanced fluid volume in the nutrition domain, impaired gas exchange in the elimination and exchange domain, ineffective breathing pattern in the activity/rest domain, deficient knowledge in the perception/cognition domain, hopelessness in the self-perception domain, impaired parenting in the role relationships domain, ineffective coping in the coping/stress tolerance domain, hyperthermia in the safety/protection domain, acute pain in the comfort domain, and delayed growth and development in the growth/development domain.

## 4. Discussion

This study examined the status of undergraduate neonatal nursing clinical practicum following the COVID-19 pandemic and identified students’ neonatal nursing training needs to develop an XR-based NICU nursing simulation training program. During the outbreak of COVID-19, 66.7% of the students had their clinical practicum off-site, which was two-fold higher than the percentage of students who had their clinical practicum on-site. A previous study that examined the status of neonatal nursing clinical practicum before the COVID-19 pandemic [24] reported rates of 47.1% and 52.9% for off-site and on-site clinical practicum, respectively. Although these data were collected from different populations, they indicate that on-site clinical practicum markedly reduced following the pandemic. Furthermore, the rate of students dissatisfied with their clinical practicum in this study was 4.4 times higher than that reported by a study prior to the pandemic [24]. This study’s results indicate that restrictions imposed on clinical practicum contributed to this situation, which coincides with existing research that explored nursing students’ opinions about COVID-19 [40].

Students who had had on-site clinical practicum demonstrated significantly higher neonatal nursing performance than those who had off-site clinical practicum. However, high-risk neonatal nursing performance did not significantly differ between the groups. Perhaps those who had on-site clinical practicum were given inadequate opportunities to provide direct patient care and primarily observed nurses because of the critical nature of newborn nurseries and NICUs. It matches prior research, which elucidates that the actual practice of skills is limited or prohibited in the NICU [2], and that students were not cleared to work independently in the NICU [4]. Concerning the method of NICU training, students preferred face-to-face and remote simulation methods for nursing practice training [5,39]. Therefore, mixed simulation training that combines face-to-face and non-face-to-face training should be used to provide environments that resemble clinical settings, thus ensuring ample opportunities to practice high-risk neonatal care.

Compared to a study conducted before the COVID-19 pandemic [4], the mean NICU training needs a score in this study was higher, while the mean NICU training experience score was lower. These results show that students’ training needs have increased because of the restrictions placed on NICU training owing to COVID-19. 

The training needs for high-risk newborn care were most common among newborn assessment, newborn care, and high-risk newborn care. Park and Ji [24] reported that nursing students wanted more training in high-risk newborn care, thus supporting this study’s findings. These results suggest that students perceived that they had been inadequately equipped with practical competency, which led to an elevated need for alternative training modalities.

The topics of NICU training with the greatest gap between needs and experiences, as indicated by the analyses in this study, were gavage tube feeding, oxygen therapy, fluid therapy, attachment promotion, and incubator management. They should be included in NICU training to promote effective training tailored to students’ specific needs. 

Regarding neonatal and family handoff training needs, students perceived a high need for handoff training, which is supported by previous research [34]. Holt, Crowe, Lynagh, and Hutcheson [40] reported that poor communication among healthcare professionals can lead to adverse patient outcomes. Studies also noted that nursing students are placed in clinical settings without being adequately prepared for handoffs [34,41]. Furthermore, researchers have stressed that unorganized nurse handoffs without an established system cause confusion and can have a detrimental impact on patients’ health [28]. Systematic NICU handoff training programs should be developed to help nursing students follow clinical practice effectively. 

After surveying specific nursing problems that required neonatal and family handoff training, the need for safety/protection was highest among the 10 nursing diagnosis areas. Students reported high training needs for the risk of fluid imbalance, impaired gas exchange, and ineffective breathing patterns, which matched the findings of prior research [34]. Hence, an XR-based neonatal care simulation handoff training program using SBAR (Situation, Background, Assessment, and Recommendation) should be developed, with a primary focus on fluid imbalance and respiratory problems observed in the NICU.

Although I attempted to enhance the generalizability of the study by enrolling nursing students from schools all over South Korea, the findings should be utilized with caution as the clinical situation may differ across schools. Therefore, I recommend additional identification of the needs of trainees and suitable modification of the training content. In addition, since this study was conducted using an online survey, the results may be biased, therefore, caution must be exercised when interpreting the results. Further, it is expected that this study’s findings will be enhanced through in-depth interviews with the participants in the future.

## 5. Conclusions

The nursing students included in this study did not have adequate opportunities for clinical practice in newborn nurseries and NICUs. Even if they are placed in clinical practicums, their primary roles are limited to observation and simple tasks. On-site clinical practicums in newborn nurseries and NICUs have been seriously restricted or suspended completely, following the COVID-19 pandemic. Newborn nurseries and NICUs require more specialized, specific, and precise care than the general ward, and the current situation hinders the utilization of clinical practicum to advance students’ NICU experiences and skills. This study’s findings showed a low rate of on-site clinical practicum in newborn nurseries or NICUs, a high need for high-risk neonatal care training, and low performance. The results highlight the critical need for alternative training modalities for clinical practicum to advance students’ high-risk neonatal care competency. Considering that students with prior handoff training perceived the need for NICU handoff training more than those without such experience, programs that foster students’ abilities to adjust to clinical practice are needed. This study provides considerable evidence for the development of systematic XR-based simulation training programs that can boost students’ clinical performance in newborn nurseries and NICUs.

## Figures and Tables

**Figure 1 ijerph-20-00344-f001:**
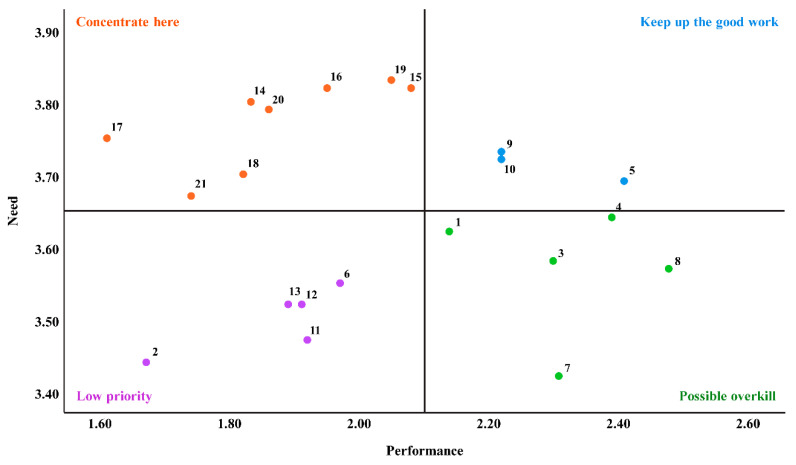
Importance-performance analysis matrix of newborn intensive care training need and experience.

**Table 1 ijerph-20-00344-t001:** General and neonatal nursing-related characteristics of participants (*n* = 132).

Characteristic	Category	*n* (%) or M (SD)
Total		132 (100.0)
Sex	Male	14 (10.6)
	Female	118 (89.4)
Satisfaction with nursing major	Dissatisfaction	6 (4.5)
	Neutral	41 (31.1)
	Satisfaction	85 (64.4)
Satisfaction with clinical practice	Dissatisfaction	23 (17.5)
	Neutral	63 (47.7)
	Satisfaction	46 (34.8)
Training sites of newborn and high-risk neonatal care	On-site (nursery/NICU)	44 (33.3)
	Off-site (campus/online)	88 (66.7)
Secure a private university hospital	Yes	19 (14.4)
	No	113 (85.6)
Simulation training experience in neonatal care	Yes	84 (63.6)
	No	48 (36.4)
Handover training experience	Yes	55 (41.7)
	No	77 (58.3)
Demand XR training about neonatal care	Yes	129 (97.7)
	No	3 (2.3)
Neonatal nursing performance	Total	2.49 ± 0.65
High-risk neonatal nursing performance	Total	2.28 ± 0.69
	Unstable vital condition	2.37 ± 0.71
	Weak response and reflex	2.55 ± 0.83
	Provision of direct treatment and nursing care	1.89 ± 0.75
	Continuous monitoring and testing	2.44 ± 0.84
Newborn intensive care training need	Total	3.67 ± 0.50
	Newborn assessment	3.59 ± 0.62
	Newborn nursing	3.63 ± 0.56
	High-risk neonatal nursing	3.77 ± 0.47
Newborn intensive care training experience	Total	2.04 ± 0.79
	Newborn assessment	2.14 ± 0.86
	Newborn nursing	2.14 ± 0.91
	High-risk neonatal nursing	1.87 ± 0.89
The necessity of handover training contents	Total	3.47 ± 0.48
	Nutrition	3.58 ± 0.57
	Elimination and exchange	3.57 ± 0.52
	Activity and rest	3.55 ± 0.57
	Perception and cognition	3.15 ± 0.96
	Self-perception	2.96 ± 0.90
	Role relationships	3.45 ± 0.66
	Coping and stress tolerance	3.21 ± 0.72
	Safety and protection	3.65 ± 0.51
	Comfort	3.47 ± 0.59
	Growth and development	3.58 ± 0.66

XR = extended reality (excluding virtual reality, augmented reality, and mixed reality technologies); M = mean; SD = standard deviation; NICU = neonatal intensive care unit.

**Table 2 ijerph-20-00344-t002:** Differences in neonatal and high-risk neonatal nursing performance, newborn intensive care training needs and experience, and handover training needs by general characteristics of participants (*n* = 132).

Characteristic	Category	Neonatal Nursing Performance		High-Risk Neonatal Nursing Performance		Newborn Intensive Care Training Need		Newborn Intensive Care Training Experience		Handover Training Need	
		M ± SD	t or F (P)	M ± SD	t or F (P)	M ± SD	t or F (P)	M ± SD	t or F (P)	M ± SD	t or F (P)
Sex	Male	2.48 ± 0.71	0.08 (0.946)	2.28 ± 0.63	0.03 (0.980)	3.53 ± 0.44	1.10 (0.274)	2.09 ± 0.77	0.24 (0.808)	3.35 ± 0.36	0.94 (0.351)
	Female	2.49 ± 0.65		2.28 ± 0.70		3.69 ± 0.50		2.03 ± 0.80		3.48 ± 0.50	
Satisfaction with nursing major	Dissatisfaction	2.23 ± 078	0.65 (0.522)	2.18 ± 0.53	0.08 (0.924)	3.65 ± 0.38	1.70 (0.187)	1.79 ± 0.51	1.76 (0.176)	3.29 ± 0.46 a	3.61 (0.030)
	Neutral	2.46 ± 0.62		2.27 ± 0.63		3.56 ± 0.62		2.22 ± 0.70		3.32 ± 0.53 b	c > a, b
	Satisfaction	2.53 ± 0.67		2.29 ± 0.73		3.73 ± 0.43		1.97 ± 0.84		3.55 ± 0.44 c	(Scheffé’s)
Satisfaction with clinical practice	Dissatisfaction	2.37 ± 0.65	0.62 (0.542)	1.96 ± 0.62 a	3.01 (0.043)	3.73 ± 0.37	0.22 (0.802)	1.79 ± 0.74	1.86 (0.160)	3.47 ± 0.40	0.29 (0.747)
	Neutral	2.50 ± 0.64		2.36 ± 0.72 b	b > a	3.68 ± 0.54		2.02 ± 0.76		3.50 ± 0.47	
	Satisfaction	2.55 ± 0.67		2.33 ± 0.65 c	(Scheffé’s)	3.64 ± 0.51		2.18 ± 0.85		3.43 ± 0.54	
Training sites of newborn and high-risk neonatal care	On-site (nursery/NICU)	2.86 ± 0.50	9.39 (<0.001)	2.44 ± 0.67	1.32 (0.189)	3.77 ± 0.33	1.09 (0.284)	2.70 ± 0.56	9.03 (<0.001)	3.48 ± 0.45	0.02 (0.987)
	Off-site (campus/online)	1.70 ± 0.68		2.24 ± 0.70		3.64 ± 0.57		1.70 ± 0.68		3.48 ± 0.51	
Secure a private university hospital	Yes	2.57 ± 0.61	0.55 (0.580)	2.19 ± 0.79	0.62 (0.538)	3.70 ± 0.34	0.30 (0.765)	2.45 ± 0.67	2.53 (0.012)	3.40 ± 0.40	0.68 (0.496)
	No	2.48 ± 0.66		2.29 ± 0.67		3.67 ± 0.52		1.97 ± 0.79		3.48 ± 0.50	
Simulation training experience about neonatal care	Yes	2.51 ± 0.66	0.31 (0.758)	2.27 ± 0.70	0.30 (0.765)	3.71 ± 0.38	1.10 (0.278)	1.95 ± 0.78	1.76 (0.081)	3.52 ± 0.41	1.58 (0.116)
	No	2.47 ± 0.66		2.30 ± 0.68		3.60 ± 0.66		2.20 ± 0.80		3.38 ± 0.59	
Demand XR training about neonatal care	Yes	2.49 ± 0.66	4.17 (0.006)	2.27 ± 0.69	2.95 (0.073)	3.67 ± 0.50	0.59 (0.555)	2.03 ± 0.80	1.38 (0.271)	3.46 ± 0.49	0.64 (0.524)
	No	2.85 ± 0.12		2.79 ± 0.29		3.84 ± 0.20		2.29 ± 0.30		3.64 ± 0.33	
Handover training experience	Yes	2.55 ± 0.64	0.87 (0.386)	2.37 ± 0.64	1.25 (0.214)	3.73 ± 0.41	1.07 (0.287)	2.07 ± 0.78	0.36 (0.717)	3.57 ± 0.46	2.07 (0.041)
	No	2.45 ± 0.66		2.22 ± 0.72		3.63 ± 0.56		2.01 ± 0.81		3.40 ± 0.49	

XR = extended reality (including virtual reality, augmented reality, and mixed reality technologies); a, b, c = different lowercase letters in lines indicate the statistical analysis results of the mean differences, and scores are summarized.

**Table 3 ijerph-20-00344-t003:** Difference between newborn intensive care training need and experience (*n* = 132).

No.	Item	Training Need	Training Experience	Gap	Paired t (*p*)	Borich Needs	Rank
		M ± SD	M ± SD	M ± SD			
Total		3.67 ± 0.50	2.04 ± 0.79	1.64 ± 0.91	20.72 (<0.001)	6.02	
Newborn assessment		3.59 ± 0.62	2.14 ± 0.86	1.44 ± 1.02	16.22 (<0.001)	5.17	3
1	Apgar scoring	3.62 ± 0.74	2.14 ± 1.03	1.49 ± 1.20	14.28 (<0.001)	5.39	15
2	Transition assessment	3.44 ± 0.79	1.67 ± 0.80	1.77 ± 1.09	18.71 (<0.001)	6.09	9
3	Neonatal behavioral status	3.58 ± 0.74	2.30 ± 1.20	1.28 ± 1.32	11.18 (<0.001)	4.58	19
4	Physical assessment	3.64 ± 0.71	2.39 ± 1.08	1.26 ± 1.22	11.84 (<0.001)	4.59	18
5	Body measurement	3.69 ± 0.68	2.41 ± 1.07	1.28 ± 1.23	12.01 (<0.001)	4.72	17
6	Gestation period assessment	3.55 ± 0.71	1.97 ± 0.1.00	1.58 ± 1.30	14.04 (<0.001)	5.61	13
Newborn care		3.63 ± 0.56	2.14 ± 0.91	1.50 ± 1.04	16.47 (<0.001)	5.45	2
7	Respiration maintenance	3.42 ± 0.68	2.31 ± 1.15	1.11 ± 1.35	9.51 (<0.001)	3.80	21
8	Maintain body temperature	3.57 ± 0.69	2.48 ± 1.14	1.08 ± 1.33	9.35 (<0.001)	3.86	20
9	Infection prevention	3.73 ± 0.62	2.22 ± 1.14	1.51 ± 1.27	13.65 (<0.001)	5.63	12
10	Nutrition supply	3.72 ± 0.60	2.22 ± 1.18	1.50 ± 1.30	13.27 (<0.001)	5.58	14
11	Bath	3.47 ± 0.78	1.92 ± 1.13	1.55 ± 1.30	13.68 (<0.001)	5.38	16
12	Attachment promotion	3.52 ± 0.76	1.91 ± 1.04	1.61 ± 1.26	14.66 (<0.001)	5.67	11
13	Discharge education	3.52 ± 0.80	1.89 ± 1.05	1.62 ± 1.25	14.96 (<0.001)	5.70	10
High-risk newborn care		3.77 ± 0.47	1.87 ± 0.89	1.90 ± 0.98	22.32 (<0.001)	7.16	1
14	Oxygen therapy	3.80 ± 0.55	1.83 ± 1.02	1.97 ± 1.11	20.49 (<0.001)	7.49	2
15	Maintain body temperature	3.82 ± 0.51	2.08 ± 1.13	1.74 ± 1.18	16.83 (<0.001)	6.65	8
16	Infection prevention	3.82 ± 0.52	1.95 ± 1.10	1.86 ± 1.18	18.19 (<0.001)	7.11	4
17	Gavage tube feeding	3.75 ± 0.60	1.61 ± 0.95	2.14 ± 1.12	22.00 (<0.001)	8.03	1
18	Incubator management	3.70 ± 0.61	1.82 ± 1.08	1.88 ± 1.29	18.21 (<0.001)	6.96	6
19	Phototherapy	3.83 ± 0.50	2.05 ± 1.20	1.78 ± 1.27	16.06 (<0.001)	6.82	7
20	Infusion therapy	3.79 ± 0.54	1.86 ± 1.09	1.93 ± 1.19	18.61 (<0.001)	7.31	3
21	Attachment promotion	3.67 ± 0.68	1.74 ± 0.98	1.93 ± 1.09	20.31 (<0.001)	7.08	5

M = mean; SD = standard deviation.

**Table 4 ijerph-20-00344-t004:** Differences in the needs for handover training according to the presence or absence of neonatal nursing clinical field practice (*n* = 132).

Nursing Diagnosis		Experienced of Handover Training (*n* = 55)	No Experienced of Handover Training (*n* = 77)	Independent *t*-Test (*p*)	Total	Rank
		M ± SD	M ± SD	t (*p*)	M ± SD	
Total		3.57 ± 0.46	3.40 ± 0.49	2.07 (0.041)	3.47 ± 0.48	
Domain 1. Nutrition		3.66 ± 0.52	3.52 ± 0.60	1.39 (0.167)	3.58 ± 0.57	2
1	Imbalanced nutrition	3.58 ± 0.66	3.42 ± 0.75	1.32 (0.189)	3.48 ± 0.72	26
2	Risk for unstable blood glucose level	3.53 ± 0.77	3.30 ± 0.89	1.54 (0.126)	3.39 ± 0.85	34
3	Risk for neonatal hyperbilirubinemia	3.75 ± 0.58	3.62 ± 0.63	1.13 (0.260)	3.67 ± 0.61	7
4	Risk for electrolyte imbalance	3.67 ± 0.61	3.64 ± 0.69	0.31 (0.754)	3.65 ± 0.65	9
5	Risk for imbalanced fluid volume	3.76 ± 0.51	3.62 ± 0.67	1.31 (0.193)	3.68 ± 0.61	5
Domain 2. Elimination and exchange		3.65 ± 0.48	3.51 ± 0.54	1.58 (0.117)	3.57 ± 0.52	4
6	Impaired urinary elimination	3.65 ± 0.55	3.56 ± 0.66	0.88 (0.379)	3.60 ± 0.62	17
7	Constipation	3.58 ± 0.63	3.31 ± 0.69	2.29 (0.023)	3.42 ± 0.68	29
8	Diarrhea	3.62 ± 0.71	3.53 ± 0.68	0.70 (0.484)	3.57 ± 0.69	21
9	Dysfunctional gastrointestinal motility	3.69 ± 0.51	3.55 ± 0.68	1.41 (0.161)	3.61 ± 0.62	15
10	Impaired gas exchange	3.71 ± 0.53	3.58 ± 0.68	1.14 (0.257)	3.64 ± 0.62	11
Domain 3. Activity/rest		3.61 ± 0.50	3.51 ± 0.61	1.05 (0.294)	3.55 ± 0.57	5
11	Activity intolerance	3.55 ± 0.69	3.34 ± 0.79	1.57 (0.118)	3.42 ± 0.75	30
12	Ineffective breathing pattern	3.73 ± 0.62	3.74 ± 0.62	0.12 (0.905)	3.73 ± 0.62	1
13	Impaired spontaneous ventilation	3.71 ± 0.53	3.58 ± 0.70	1.12 (0.266)	3.64 ± 0.63	12
14	Risk for unstable blood pressure	3.47 ± 0.74	3.35 ± 0.86	0.85 (0.395)	3.40 ± 0.81	32
15	Ineffective peripheral tissue perfusion	3.65 ± 0.55	3.52 ± 0.72	1.17 (0.244)	3.58 ± 0.66	18
16	Dysfunctional ventilator weaning response	3.56 ± 0.69	3.51 ± 0.72	0.46 (0.647)	3.53 ± 0.70	24
Domain 4. Perception/cognition		3.40 ± 0.83	2.97 ± 1.01	2.56 (0.011)	3.15 ± 0.96	9
17	Deficient knowledge	3.40 ± 0.83	2.97 ± 1.01	2.56 (0.011)	3.15 ± 0.96	42
Domain 5. Self-perception		3.19 ± 0.81	2.80 ± 0.93	2.51 (0.013)	2.96 ± 0.90	10
18	Hopelessness	3.25 ± 0.80	2.78 ± 0.99	2.93 (0.004)	2.98 ± 0.95	45
19	Situational low self-esteem	3.13 ± 0.88	2.82 ± 0.94	1.91 (0.059)	2.95 ± 0.93	46
Domain 6. Role relationships		3.54 ± 0.57	3.38 ± 0.71	1.41 (0.162)	3.45 ± 0.66	7
20	Caregiver role strain	3.55 ± 0.63	3.34 ± 0.81	1.59 (0.113)	3.42 ± 0.74	31
21	Impaired parenting	3.60 ± 0.63	3.51 ± 0.75	0.75 (0.453)	3.55 ± 0.70	22
22	Dysfunctional family processes	3.60 ± 0.60	3.43 ± 0.80	1.34 (0.182)	3.50 ± 0.73	25
23	Readiness for enhanced family processes	3.49 ± 0.72	3.39 ± 0.73	0.79 (0.429)	3.43 ± 0.72	27
24	Parental role conflict	3.47 ± 0.72	3.23 ± 0.87	1.67 (0.098)	3.33 ± 0.82	36
Domain 7. Coping/stress tolerance		3.42 ± 0.68	3.06 ± 0.71	2.87 (0.005)	3.21 ± 0.72	8
25	Anxiety	3.49 ± 0.69	3.13 ± 0.80	2.70 (0.008)	3.28 ± 0.78	37
26	Ineffective coping	3.45 ± 0.77	3.29 ± 0.78	1.24 (0.217)	3.36 ± 0.77	35
27	Death anxiety	3.45 ± 0.86	2.99 ± 0.88	3.04 (0.003)	3.18 ± 0.90	40
28	Fear	3.40 ± 0.76	3.03 ± 0.81	2.68 (0.008)	3.18 ± 0.81	41
29	Grieving	3.40 ± 0.78	3.12 ± 0.84	1.96 (0.052)	3.23 ± 0.83	39
30	Powerlessness	3.40 ± 0.81	2.91 ± 0.88	3.28 (0.001)	3.11 ± 0.88	43
31	Impaired resilience	3.38 ± 0.81	2.90 ± 0.90	3.20 (0.002)	3.10 ± 0.89	44
32	Stress overload	3.36 ± 0.78	3.17 ± 0.89	1.30 (0.196)	3.25 ± 0.85	38
Domain 8. Safety/protection		3.68 ± 0.45	3.63 ± 0.54	0.57 (0.572)	3.65 ± 0.51	1
33	Risk for infection	3.67 ± 0.64	3.64 ± 0.63	0.33 (0.745)	3.65 ± 0.63	10
34	Ineffective airway clearance	3.71 ± 0.53	3.66 ± 0.62	0.45 (0.652)	3.68 ± 0.58	6
35	Risk for aspiration	3.67 ± 0.64	3.71 ± 0.63	0.37 (0.710)	3.70 ± 0.63	3
36	Risk for bleeding	3.69 ± 0.51	3.55 ± 0.72	1.29 (0.199)	3.61 ± 0.64	16
37	Risk for falls	3.67 ± 0.61	3.61 ± 0.71	0.53 (0.599)	3.64 ± 0.67	13
38	Risk for thermal injury	3.58 ± 0.60	3.51 ± 0.70	0.65 (0.519)	3.54 ± 0.66	23
39	Impaired skin integrity	3.71 ± 0.50	3.64 ± 0.69	0.67 (0.504)	3.67 ± 0.61	8
40	Hyperthermia	3.69 ± 0.61	3.73 ± 0.64	0.33 (0.743)	3.71 ± 0.62	2
41	Hypothermia	3.67 ± 0.64	3.60 ± 0.67	0.65 (0.519)	3.63 ± 0.66	14
42	Ineffective thermoregulation	3.75 ± 0.55	3.68 ± 0.64	0.66 (0.511)	3.70 ± 0.60	4
Domain 9. Comfort		3.56 ± 0.55	3.41 ± 0.61	1.42 (0.159)	3.47 ± 0.59	6
43	Impaired comfort	3.49 ± 0.66	3.39 ± 0.71	0.83 (0.408)	3.43 ± 0.69	28
44	Nausea	3.51 ± 0.69	3.32 ± 0.77	1.42 (0.159)	3.40 ± 0.74	33
45	Acute pain	3.67 ± 0.58	3.52 ± 0.72	1.36 (0.178)	3.58 ± 0.67	19
Domain 10. Growth/development		3.67 ± 0.58	3.51 ± 0.70	1.49 (0.139)	3.58 ± 0.66	3
46	Delayed growth and development	3.67 ± 0.58	3.51 ± 0.70	1.49 (0.139)	3.58 ± 0.66	20

M = mean; SD = standard deviation.

## Data Availability

Not applicable.

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
