# Peer review of "Impact of the COVID-19 Pandemic on Neonatal Nursing Practicum and Extended Reality Simulation Training Needs: A Descriptive and Cross-Sectional Study"

_ijerph, 2022, doi:10.3390/ijerph20010344_

Round 1
Reviewer 1 Report
You write as a theoretical framework seems to correspond to the objectives of the study. I would like to know more information published about the xr in other universities or at the international scientific community, this is a theoretical framework.Author Response
Responses attached.

Reviewer 2 Report
Interesting theme and well contextualized in the impact of Covid-19 on the teaching-learning process in nursing training. The purpose for which it is proposed must be continued.
Line 207-208: "high-risk neonatal nursing performance (t= 2.95, p =.073) were significantly" has no statistical significance, p must be less than .05 and here it is .073. The mention of this result should be corrected.
Page 6: 24 "Parental role conflict" rectify 1.67? (.098), the question mark must not be included.
Author Response
Responses attached

Reviewer 3 Report
Considering scientific writing precepts, I suggest that the description made in “theoretical framework” be relocated to the methods section, as it portrays how the study was structured.
Method:
A question about inclusion criteria in the sample: were all students included, but did all of them respond fully to the research instrument?
The article states that “Participants were pre-licensed nursing students enrolled in an undergraduate nursing program in South Korea”, however, I see the need to detail this information, as nursing students from all schools in Korea South participated. How many schools were considered?
In the instruments section: the sample characterization variables were not described (although they are shown in Table 1).
Discussion:
Doubt: The article states that the data were collected after the end of the COVID-19 pandemic, but only in December 2022 did the WHO signal this statement. Which references were the authors based on to consider the pandemic over? (A suggestion: review the wording of the text on this aspect).
It is necessary to include in the limits of the study those related to the biases of an online survey.
Conclusion:
In the first sentence presented, align with the results of the study, which are sufficient to state that “The nursing students in the present study did not have adequate opportunities for clinical practice in newborn nurseries and NICUs”.
Congratulations!! I wish you a good work.
Author Response
Responses attached
